



# Identifying climate variables that interchange with volcanic eruptions as cooling forces during the Common Era's ice ages.

Seip[1], Knut.Lehre. Øyvind Grøn[1], Hui Wang[2]

[1]Faculty of Technology, Art and design, Oslo Metropolitan University, Oslo, N-0130, Norway
[2] NOAA/NWS/NCEP/Climate Prediction Center, 5830 University Research Court, NCWCP, College Park, MD 20740, USA

*Correspondence to*: Knut L. Seip (knut.lehre.seip@oslomet.no)

**Abstract.** Volcanism is known to be an instigating factor for the Late Antique Little Ice Age (LALIA, 536-660) and the Little Ice Age (LIA, 1250 – 1850), but little is known about when the effect of volcanism ends, and which other mechanisms prolong a cold period that includes the ice-ages' cold periods, but also continued periods with persistent cooling. Here we show, with
a high-resolution lead-lag method, where the stratospheric aerosol optical depth (SAOD) generated by volcanic emissions ceases to precede the Northern Hemisphere summer temperature (NHST). We find that five climate mechanisms cool the Northern Hemisphere (percentage time in parentheses): SAOD (51%), total solar irradiance (TSI, 2%), the North Atlantic oscillation (NAO, 11%), the interdecadal Pacific oscillation (IPO, 28%) and $CO_2$ (16%). The last four variables overlap, and altogether the five climate variables cover 89 % of the cold period that includes LALIA and LIA. In contrast, we find an
increase in atmospheric $CO_2$ over a brief period just after large volcanic eruptions. During the cold period, the five variables lead NHST, are in a cooling mode, and have sufficient strength to cool the Northern Hemisphere.

## 1 Introduction

A causal role of volcanic eruptions for low temperatures in the Northern Hemisphere is suggested in many studies. Causal
inferences may be expressed based on reasonable assumptions, Shi et al. (2022), model simulations, Slawinska and Robock (2018, p. 2163),  van Dijk et al. (2023), and on a significant negative correlation between the Northern Hemisphere summer temperature (NHST) and volcanism, the latter expressed as  total stratospheric aerosol optical depth (SAOD, Buntgen et al. (2020, p.6)). Recently, Marshall et al. (2022) gave a summary of volcanic effects on the climate.

Volcanic eruptions instigate cold periods, but other mechanisms may maintain or continue the cooling. However, to our
knowledge, few studies examine the timings when the effect of volcanism ceases, and other mechanisms continue the cooling effect. Knowledge of those timings (the *whens*) may increase the possibility of finding the variables (the *which*) that continue the cooling and the processes (the *whys*) that explain the incidence of ice ages ($\approx$ 536 – 660; 1250 – 1850) in the Northern Hemisphere.

As a measure of the volcanic impact on temperature, the SAOD is used  (Ridley et al., 2014; Sato et al., 1993). It is based
on measurements of stratospheric aerosol content. However, water vapor, (Kroll 2021), and ash, Wells et al. (2023) also affect





the temperature. Additional factors that effect the temperature are the height the erupted substances reach with respect to the troposphere, and the latitude of the volcanoes (Aubry et al., 2020; Jenkins et al., 2023; Marshall et al., 2022).

A variable which could extend the ice age is a low value of $CO_2$ in the atmosphere, e.g., resulting from changes in ocean variability,  (Jones & Cox, 2001) or from changes in net primary production (Goosse et al., 2006; Jones & Cox, 2001; van Dijk

et al., 2023). Additional mechanisms that could extend  low temperatures are changes in cloud albedo and cloud area, Arctic Sea ice, vegetation cover, ocean circulation (Dee & Steiger, 2022; Slawinska & Robock, 2018), and solar forcing, Steinhilber et al. (2009). The North Atlantic Oscillation (NAO) and the Pacific multidecadal variability (PMV)  are both suggested to be instigated by the cooling resulting from volcanic activities, van Dijk et al. (2023) and Sun et al. (2022), respectively.

Here we examine the leading role that volcanism played in NHST, and thus the potential causal effect that volcanism had

on the low temperatures during the Late Antique Little Ice Age (LALIA) and the Little Ice Age (LIA). As alternative candidates to decrease and maintain cold temperatures we choose total solar irradiance, (TSI), the North Atlantic oscillation (NAO), the interdecadal Pacific oscillation (IPO) and carbon dioxide, $CO_2$. We disentangle the observed series for NHST to identify low and high frequency components. The low frequency component would correspond to changes in climate – and therefore has centennial to millennial cycle periods - and causation would refer to mechanisms that act on long time scales. On decadal time

scales other mechanisms are candidates. We disentangle the series by first applying LOESS smoothing (see below) to the observed series, and thereafter by applying the high-resolution lead-lag (HRLL) method to the low and high frequency time series of SAOD, NHST and to the alternative climate series.

## 1.1 Hypotheses

For the lead-lag (LL) analysis, we use a high-resolution lead-lag  (HRLL) method, Seip et al. (2018) that identify LL

relations over very short time windows (three synoptic observations in paired time series, nine synoptic observations are sufficient for calculating confidence intervals). The HRLL method also calculates running cycle periods and phase shifts. We first hypothesize, **H1**, that after major volcanic eruptions, SAOD will lead NHST, and temperatures will decrease during periods in the Late Antique Little Ice Age, LALIA, and during periods in the Little Ice Age, LIA. This will cause short cycle periods in SAOD (corresponding to the rapid cooling) to instigate long cycle periods in NHST. *Second*, we hypothesize, **H2,**

that the dates when the leading role of SAOD to NHST ceases, other variables will replace volcanic eruptions as cooling factors. *Third*, we hypothesize, **H3**, that volcanic eruptions will instigate variables that preserve and continue the cold temperatures during the ice ages.

Throughout the manuscript we will for convenience use the term "cycle period" (CP) although changes in period lengths may indicate that the variabilities are not strictly cyclic. Furthermore,  time series may be superpositions of several cyclic

series generated by different mechanisms, (Seip & Pleym, 2000),  including dynamic chaos,  (Sugihara & May, 1990). Correctly disentangling such series is therefore an important task in climate studies (Fang et al., 2021; Wills et al., 2018).



The LL hypothesis comes with a caveat; if two variables impact the temperature, the second variable may distort the temperature series so that the first series appears to lag the temperature series even if it is leading. However, it is not symmetric, if a leading series is significantly leading, there is a low probability that the second variable can distort the relation for a longer

period (≈ 10 years).

Our study shows for the first time high-resolution lead-lag relations between climate variables during the Common Era. We also, for the first time, add cycle period series describing changes in cycle periods as "sister" series to all series. We do not intend to give detailed descriptions of mechanisms that govern the relations between climate variables. We believe this can best be done together with modeling studies. The present study would then provide independent information on relations

between climate variables that can be used to evaluate modeling results.

In the rest of the manuscript we describe the data used in Section 2. The method is described in Section 3 with emphasis on the HRLL method. Section 4 shows the results that are discussed in Section 5. Section 6 concludes.

## 2. Data

In this section we describe the climate series we use in the study. We also outline how we identify time windows for the periods with decreasing and cold temperatures. The periods include the LALIA and the LIA. For convenience, we do not add the term "year" before numerical year designations.

*Northern Hemisphere summer temperatures*. We use Northern Hemisphere summer temperature series, NHST, during the Common Era from  Büntgen (2020). The data series was supplied by Ulf Büntgen and is a relatively new version of

temperatures calculated for  the Northern Hemisphere (Anchukaitis & Smerdon, 2022).  However, temperature series may be biased due to inferences from proxy records or shifts in the temperature measure instruments during the early instrumental period 1750-1900  (Buntgen et al., 2020; Böhm et al., 2010).

*Volcanic eruptions.* The stratospheric aerosol optical depth, SAOD, expresses the cooling potential of volcanic eruptions. We compare two estimates of SAOD. The first one, $SAOD_B$, is a series for the 500 nm from 30ºN to 90ºN used by Buntgen et

al. (2020) and supplied by Fredrik Charpentier Ljungqvist. The series have references to Sigl et al. (2015) and Jungclaus et al. (2017). The second series, $SAOD_S$, is an adjusted series that were calibrated based on ice core data from Antarctica and Greenland (Toohey et al., 2016)**.** The two series differ, $\Delta SAOD = SAOD_B - SAOD_S$, with an average of < 1% of the means. The largest difference for any year and the average is 40%. However, when we calculate the LL relation between $SAOD_B$, $SAOD_S$ and NHST, the LL relations are almost identical. In the following, we use the $SAOD_B$ series in the calculations and

drop the B suffix. All series are used in LOESS (0.05), LOESS (0.1), LOESS (0.2) and LOESS(0.3) smoothed formats. LOESS smoothing is discussed in the method section below.

*The volcanoes*. Among the 25 largest volcanic eruptions during the last 2500 years there are four volcanic eruptions during the LALIA and eleven eruptions during the LIA, Table 1, Sigl et al. (2015, extended data, Table 4).

*Definitions of cold periods*. Here we define ice age periods as time windows where the temperature of the LOESS(0.2)

smoothed NHST series is i) lower than $T_{ave}$ -0.5 standard deviation (SD) and ii) nine-year periods have accumulated





temperatures less than $T^\times$ (see below). The latter requirement eliminates non-persistent time windows with low temperatures. Since volcanic eruptions tend to decrease temperatures abruptly, we also identified periods where the $T_t$-$T_{t-1}$ decrease is more than the $\Delta T_{ave} - 1$ SD.

## 2.1 Periods during the Common Era

We divided the Common Era into two periods, one from 1 to 1000 and one from 1001 to 2017. The average temperature for the first period of NHST was -0.18°C ± 0.42 °C and for the second period of NHST it was -0.37 °C ± 0.44°C, thus, in the last millennial, temperatures were on average 0.19°C lower than in the first millennial.

*The Roman warm* period is often defined as the period from 250 BC to 400 AD with peaks during the summer warming 280-287 (Buntgen et al., 2020).

*The Late Antique Little Ice Age, LALIA*. The most frequently used time window for the LALIA is from *536 to 660*. There is no firm definition of the period for the LALIA, and we find that there are four periods with cold temperatures, and two periods with decreasing temperatures, and together they form a semi continuous period with decreasing and cold periods from 245 to 884, Table 1.

*The medieval period* (MED) is often defined as the period between 476 AD (the fall of the Western Roman Empire) and

1453 AD (the fall of Constantinople). There were peaks in summer warming between 1020 and 1031 (Buntgen et al., 2020).

*The Little Ice Age, LIA*. We used the same technique as described above for the LALIA to identify periods with persistent low temperatures and periods with decreasing temperatures. The LIA is often defined as the period from 1250 to 1850. Screening for non-persistent low temperatures with $T^\times = 4.5$°C, we found two periods with cold temperatures and two periods with decreasing temperatures. Cold and decreasing temperatures last from 1039 to 1824, but with a 150-year long warm spell

from 1330 to 1480.

## 2.2 Candidate cooling variables

*Total solar insolation* (TSI) expresses solar intensity and is represented by an index for sunspots. We used a series for sunspots from 1700 to 2020 supplied by Clette et al. (2015). These authors calculated 11 years running mean, and we adopted that technique to obtain the series for the present study. https://www.sidc.be/SILSO/datafiles.

*The North Atlantic oscillation (NAO)* is a frequently used candidate for a mechanism that would interchange with SAOD to continue the ice ages. It is defined as the sea level pressure (SLP) difference between the Iberian Peninsula and Iceland, $NAO = SLP_{IB} - SLP_{Ice}$, Hernández et al. (2020). If SLP is high on the Iberian Peninsula and low in Iceland, NAO is designated as NAO+. The NAO+ tends to cause warmer and wetter climate over the Northern Europe, so we will use NAO- in the calculations. The data were retrieved from https ://doi.panga ea.de/10.1594/PANGA EA.92191 6.






Table 1. Ice age characteristics.

| | Characteristics | The Antique Little Ice Age | The Little Ice Age |
|---|---|---|---|
| 1 | Formal period, years | 536-660 | 1250-1850 |
| 2 | Cold periods, years | 1-41, 486-626, 765-884, 980-1000 | 1201-1330, 1541-1824 |
| 3 | Decreasing temperatures | 245-563, 685-844 | 1039 – 1247, 1480 - 1622 |
| 4 | Average cold and decreasing temp. periods, years | 644 | 638 |
| 5 | Large volcanic eruptions, years, (global forcing, Wm$^{-2}$, in parentheses) [1] | 536 (-11.3), 540 (-19.1), 574 (-14.5), 636 (-8.2) | 1230 (-15.9), 1258 (-32.8), 1286( -9.7), 1345 (–9.4), 1458 (-20.5), 1601( -11.6), 1641 (-11.8), 1695 – (-10.2), 1783 (-15.5), 1809 (- 12), 1815 (-17.1) |
| 6 | SAOD volume | 9.6 | 13 |
| 7 | Years SAOD leads NHST | 1-30, 116-179, 327-379, 438-529, 714-773 | 1184-1417, 1456-1469, 1508-1540, 1619-1758. 1822-1865, 1938-2017 |
| 8 | Average lead periods, all years | 59.8 ± 22.3 Total leading 300 | 87.1± 83.4 Total leading 645 |
| | Total leading dec. &cold | 236 | 423 |
| 9 | Average between peak SAOD periods, years | 130± 77, n=12 | 64.3±46.5 |
| 10 | "Void" periods [2] | 31-41, 245-325, 380-437, 530-626, 685-713, 774-844, 980-1000 | 1039-1183, 1480-1507, 1541- 1618, 1759-1821 |
| 11 | Average between end and begin of leading relations. Decr. &cold, years. | 75.2 ± 32.5, n = 7 Total "void" 376 | 78.5±49.0, n = 4 Total "void," 314 |

(1) The list for large volcanic eruptions is from Sigl et al. (2015). We use eruptions with largest global forcing during the period 1230 to 1850.
(2) The term "void" is used for time window where the Northern Hemisphere summer temperature, NHST, is decreasing or cold, but volcanic eruptions do not affect the temperature.

*The Interdecadal Pacific Oscillation (IPO)*. The IPO is observed temperature changes in the surface waters of the Pacific. We use a version obtained for the years 1 to 2020 from Vance et al. (2022). It covers the regions 25°N–45°N, 140°E–145°W, 10°S–10°N, 170°E–90°W and 50°S–15°S, 150°E–160°W  https://psl.noaa.gov/data/timeseries/IPOTPI/.  A positive phase is characterized by a cooler than average northern Pacific and a warmer than average tropical Pacific (Meehl et al., 2016). Since we compare the IPO to the NHST, we will require IPO to be positive, IPO+. The data were downloaded from Australian Antarctic Data Centre :(https://data.aad.gov.au), https://doi.org/10.26179/5zm0-v192.





*Carbon dioxide* (CO$_2$) affects the global climate and we used a series provided by Neukom et al. (2019). However, since about 1850 CO$_2$ concentrations increase dramatically by anthropogenic factors. When we centered and normalized the data to unit standard deviation, we only used the series between years 1 and 1850. The data were downloaded from: http://www.nature.com/authors/editorial_policies/license.html#terms

## 3. METHOD


Establishing causality relations between variables is a major challenge in the present study. We use five "clues" for causality relations. We outline the tools in the context of ice age temperatures: i) Ordinary linear regressions (OLR) and principal component analysis (PCA) show if the candidate causal variable varies as the candidate target variable, ii) if a candidate variable influences the target, it must lead the target. Thus, we applied a high-resolution lead-lag, HRLL, algorithm

to the candidate cause/ target pairs. iii) We examine if dated volcanic eruptions correspond to particular events in the time series, iv) We examined if there were established physical or chemical mechanisms that could explain cause and target effects. v) In the present study, if a variable contributes to the ice ages, it must be in a mode that contributes to a low temperature in the Northern Hemisphere. We identified low and high values of the variables separately for the first and the second millennium by subtracting the corresponding average values for the periods 1-1000 and 1001 - 2000. We applied this procedure to the

observed series and to the "sister" series for the series cycle periods. The rationale for examining the cycle periods is that a change in a variable that lasts for a certain period (e.g., a volcanic eruption) should be reflected as a change in associated variables.

The main technique for responding to hypotheses **H1** and **H3** (the cooling hypotheses) is therefore first to identify relevant characteristics for the time series and second to construct a decision "three", here in the format of a "0"," 1" matrix with ≈ 2000

rows and 7 columns. The matrix combine the information we have obtained for pairs of time series. Each row in the matrix corresponds to a year during the Common Era and each column represents criteria that determine whether a climate variable contributes to decreasing or maintaining cold climate during ice ages (our extended definition of a cold climate includes persistent decreasing temperatures). Criteria (C1) identifies ice-ages, C2 identifies years where SAOD leads NHST, C3 identifies ice age years where SAOD does not lead NHST, C4 identifies TSI- values (low TSI), C5 identifies NAO-, C6

identifies IPO+, C7 identify CO$_2$-. Criteria C1 was described in the Section 2 on data, Criteria C2 and C3 are described below. Criteria C4 to C7 are straight forward use of climate time series. We respond to hypothesis **H2** by calculating cycle periods with a high-resolution cycle period (HRCP) method.

### 3.1 The high-resolution lead-lag method

The high-resolution lead-lag, HRLL, method we use here is based on the duality between a time series presentation and a
phase plot presentation of paired, x(t) and y(t) cyclic, non- gaussian series where one series is a candidate for effects on the





other. The method is described in detail in Seip et al. (2018). Here we give a summary based on two basic equations used to identify i) lead-lag (LL) strength and ii) cycle periods, λ, Fig. 1.

To identify LL patterns, we depict the potential causal and effect series in a phase plot. In this plot, we can determine the lead-lag pattern based on three paired consecutive observations (but the LL relations are most easily seen by comparing peaks or troughs in the two series). Figure 1a shows that if one time series peaks before another (blue and blacks), then the trajectories rotate clockwise (blue and black in Fig. 1b, black on the x-axis). If one time series peaks after another (black and red), then the trajectories rotate counterclockwise (black and red in Fig. 1b, black on the x-axis).

When persistent cycles are completed in the time series representation for the paired series, the trajectories in phase plot representation form a closed curve. To identify closed curves, we use Eq. (1) to calculate rotational angles in the phase plot.[1]

$$\theta = sign(\overline{v}_1 \times \overline{v}_2) \cdot A\,cos\left(\frac{\overline{v}_1 \cdot \overline{v}_2}{|\overline{v}_1| \cdot |\overline{v}_2|}\right) \tag{1}$$

Where $\overline{v}_1$ and $\overline{v}_2$ are vectors between three consecutive observations in the phase plot.

The LL strength is defined as Eq. (2),

$$LL = (N_{pos} - N_{neg})\,/(N_{pos} + N_{neg}) \tag{2}$$

where $N_{pos}$ and $N_{neg}$ is the sum of signs of respectively positive and negative angles θ over a period n.

With an interval of nine consecutive time steps, a comparison with paired random numbers shows that LL strength is significant when LL > + 0.32 or LL < - 0.32. However, when the raw series are smoothed, the confidence intervals do not strictly apply. The significance of LL can be used to assess the significance of the cycle period, λ.

---

[1] It can be implemented in Excel format: With $v_1$ = (A1, A2, A3) and $v_2$ = (B1, B2, B3) in an Excel spread sheet, the angle is calculated by pasting the following Excel expression into C2: =SIGN((A2-A1)*(B3-B2)-(B2-B1)*(A3-A2))*ACOS(((A2-A1)*(A3-A2) + (B2-B1)*(B3-B2))/(SQRT((A2-A1)^2+(B2-B1)^2)*SQRT((A3-A2)^2+(B3-B2)^2))).



**Figure 1**. How to know which series is leading which. a) Three series, the blue series leads the black series because it peaks before the black series, and the red series trails the black series because it peaks after the black series. b) Phase plot with sine (0.1t) on the x-axis and sine(0.1t+7): blue, and sine (0.1t-7): red, on the y-axis. c) Application of the HRLL methods cycle period identification mechanism to a synthetic series y=SIN ((0,05-SIN (0,005t) ×0,01)* t) . (blue line). The cumulative cycle period technique gives the red saw toot pattern so that the distance between the peaks corresponds to cycle periods in the blue curve. The black curve is a LOESS(0.3) smoothed version of the tracing method curve. d) Two series from bottom: known and unknown causes, Top: observed (red) and anticipated (yellow) observations. The cause leads the anticipated observations.



### 3.2 Cycle periods.

When the sum of angles, $\Sigma\theta$ equals $2\Pi$ the curves are closed and the number of time steps correspond to the common cycle
period, $\lambda$, the "cumulative angle" method. We also calculate running angles by tracing the curve in the phase plot until it closes, the "tracing method." The cycle period for single series can be found by letting a second series be a copy of the first but shifted more than 7 time steps and less than $\lambda/4$ relative to the first. Figure 1c shows cycle identification of a synthetic curve with changing cycle periods (lower blue curve), the red see-saw curve above it shows how new time steps are added until the trajectory of the curve pair (original x-axis) and its shifted copy (y-axis) closes. The thick red curve shows a LOESS(0.3)
smoothed version of the cycle periods obtained with the trace method. The HRLL method also allows us to estimate phase shifts between paired cycle series, but this will not be used here. We also identify cycle periods for single series applying Power spectral analysis, (PSA), to the series.

### 3.3 LOESS smoothing.

LOESS smoothing is a common smoothing algorithm available in almost all statistical packages. It has two parameters, (f)
that shows the fraction of the time series used as a moving window and (p) that is the polynomial degree used for interpolation. Since we always use p = 2, we use the nomenclature LOESS(f).

With knowledge of LL relations and phase shifts it is possible to compare causal and target cyclic time series as if there were no time lag between cause and effect.

### 3.4 Comment on the HRLL method

*One known and one unknown causal variable*. With two variables that both are causal to the target variable, the sign of the LL relation between the cause (known) and the target may cease to show anticipated results (that the cause is leading the target). In our study, the two causes are the effect of volcanic eruptions (known) and the mechanisms that follow the eruptions, either because of cooling by an unknown variable or instigated by the eruption. Or, third, occurring by chance. Figure 1d shows the results for a synthetic example. The two causes are represented by two sine functions with angular frequencies, $\omega = 0.2$
and $\omega = 0.25$. The anticipated result of the known cause is a displaced sine function, say, $0.785 \approx \lambda/4$ after the causes (yellow sine curve). However, the observed target curve resulting from the two causes with weights 1.0 and 0.7 respectively, is shown as the thick red curve. Calculating LL relations between the known cause and the observed target curve gives the "anticipated" LL relations 73 % of the time. If the unknown cause contributes less than 40% of the causal effect, the leading relation of the known causal variable is correct 100 % of the time.

*How to know that LL relations and cycle periods are reasonable*? All time series will have some degree of uncertainty added. Two stochastic time series will show cycle periods 7 time steps or shorter with a probability $p > 0.05$ (Seip & Wang, 2022). *Second*, stochastic time series will typically show significant LL relations less than 50% of the time. *Third*, we examine





if peaks in the leading series are closer to the peak following it than to the peak preceding it in the target series. *Fourth*, we compare estimated cycle periods and the number of cycles to the cycle periods we can visually identify in the raw and the
smoothed series.

The LL method is simple and is implemented in one spread sheet. The data, the method and all essential calculations are available from the first author. A few calculations, e.g., LOESS smoothing, PSA, and PCA, are made in SigmaPlot.

## 4. RESULTS

We first show results for the volcanic eruptions, the SAOD and the NHST. Second, we show results for the four candidate
drivers of cold temperatures: TSI, NAO, IPO and $CO_2$. Third, we examine if volcanism and the low temperatures caused by the eruptions instigate the four candidate drivers that continue or preserve the cold climate.

### 4.1 Volcanism and temperature

Two versions of the SAOD and the NHST series are shown in Fig. 2a. The upper curve shows the SAOD series slightly LOESS(0.05) smoothed. The middle two curves show the raw SAOD values (transparent blue curve) and the LOESS(0.2)
smoothed version (bold blue curve). The lower two series show the raw NHST values (transparent red curve) and the LOESS(0.2) smoothed version (bold, red curve). The dropdown lines show four peaks in the SAOD series.

The two LOESS (0.2) smoothed series are counter cyclic:

$$SAOD = - 0.596 \text{ NHST} + 0.00, R = - 0.56, p < 0.001, n = 2022 \qquad (3)$$


The curves in Fig. 2c and d show the LOESS(0.2) smoothed series of SAOD and an estimate of the duration of the ice ages. The lower blue horizontal lines show periods with low temperature and the horizontal upper red lines show periods with decreasing temperature. The red bars show when SAOD is leading NHST. Altogether, there are seven periods lasting on average $75 \pm 32$ years (range 10 to 96 years) before and during the LALIA where other variables decrease or maintain low
temperatures. During the LIA there are four periods where other variables decrease or maintain low temperature, Table 1. The periods last on average $79 \pm 49$ years (range 28 to 145 years)

The PSD graphs for the SAOD (yellow) and the NHST (blue) are shown in Fig. 2b. The dropdown lines show that the mode values for the distribution of cycle periods identified with the LL method (shown below) correspond roughly to cycle periods identified by the PSD. Both series, separately and in common, show cycle periods of 80 to 90 years (but the periods
are not significant).










**Figure 2** Effects of volcanism on the LALIA (536- 660) and the LIA (1250- 1850). a) SAOD, and NHST, raw data and LOESS (0.05) and

LOESS (0.2) smoothed. b) Power spectral density for SAOD and NHST series. The dropdown lines show averages cycle periods for SAOD

(triangle at 61 years), NHST (inverted triangles at 63 and 106 years) and for their common cycle period (filled circle at 91 years). c) The

years 1 to 1000, including LALIA: SAOD (black lines and blue bars) compared to periods where SAOD leads NHST (transparent red bars).

Horizontal green lines show periods with decreasing temperature, red bars show the ice ages. d) The years 1001 to 2022 including LIA: same

as for (c ). e) Cycle periods for the Norther Hemisphere Summer temperatures, NHST (blue) and stratospheric aerosol optical depth, SAOD

(red). Series LOESS(0.05) and LOESS(0.2)  smoothed. Droplines show large volcanic eruptions. d) Histogram for cycle periods for NHST

(red) and SAOD (green).

The LL method allows us to calculate running cycle periods. The cycle periods calculated with the *trace method* for slightly

LOESS (0.05) smoothed series and for the strongly LOESS(0.2) smoothed series of the NHST and the SAOD series are shown

in Fig. 2e. The *cumulative angle* method identifies 19 cycle periods with an average cycle period of $79 \pm 21$ years for the

NHST and 15 cycles with an average of   98 ±54 years for the SAOD. The OLR for the CP(SAOD) / CP(NHST), both

LOESS(0.2) smoothed are significant, $p < 0.001$, but the regression coefficient is small, R = - 0.07. The droplines show the

years for large volcanic eruptions. Qualitatively, it may look as if volcanic eruptions instigate longer cycle periods than the

average for NHST.

The histograms in Fig. 2f compare the distribution of cycle periods for the SAOD and NHST series. The SAOD shows two

peaks, one at $\approx 60$ year and one at $\approx 200$ year. The first peak corresponds with a peak for the NHST at $\approx 63$ years, which is

close to a peak at 61 years in the PSD graph in Fig. 2b.

**4.2 Alternative drivers of cold climate (TSI, NAO, IPO and $CO_2$,)**

We evaluated four global variables as candidates to continue decreasing or maintaining low temperatures instigated by

volcanism. These were TSI, NAO, IPO and $CO_2$. We first show the LL relations between the variables and NHST and thereafter

their cycle periods.

**4.2.1 Lead-lag relations and cooling mode**

In Fig. 3a to d we show the time series for NHST and the alternative series. We draw two horizontal lines at the top of the

graph, the lower one showing periods with decreasing or cold temperatures (blue line) and the upper one showing where the

candidate series is a possible driver of low temperatures (yellow line).  The possible drivers must be leading NHST, and they

must have been shown to lead to significant low temperatures in the Northern Hemisphere. Figure 3e shows a PCA loading





plot for the alternative variables in relation to the "voids". The "Void" variable means that the temperature is decreasing or
cold, but that volcanic eruptions are not the cause of the cooling. All variables were coded "1" or "0" (since PCA require data
that are not numerical identical, we added a small random component to all data.) For the alternative variables a "1" means
that it is leading NHST and it is in a cooling mode. The result shows that IPO are closely associated with VOID and an OLR
shows

$$IPO = 0.436 \text{ VOID} + 0.454, R = 0.44, p < 0.001, n = 1831 \qquad (4)$$


     To see when NAO and IPO were leading or lagging SAOD, we calculated the series for the LL (NAO, SAOD) and LL(IPO,
SAOD). The results are shown in Fig. 3f. NAO and IPO are leading SAOD 68% and 63 % of the time, respectively.

     There are 623 years where NHST is decreasing or cold, but where SAOD no longer decreases or maintains the temperature.
During these years, other climate variables contribute to low temperatures: TSI for 23 years (2 % of the whole period of 1282
years), NAO for 139 years (11%), IPO for 356 years (28%) and $CO_2$ for 207 years (16%). However, the four variables overlap
for some years, so altogether, the four variables we have identified contribute to decreasing and cold NHST for 480 years
(38%). Thus, the SAOD (51 %) and the four climate variables cover 89% of the period with decreasing and cold temperatures.




a)



b)



c)


d)



e)


f)







**Figure 3** Variables that may interchange with volcanic eruptions in cooling and maintaining cold (ice age) temperatures. a) TSI, b) Carbon dioxide, $CO_2$, c) NAO, d) IPO, PCA loading plot for the four alternative climate time series, and the time series for "void" periods. TSI is total solar irradiance (sunspots), $CO_2$ is carbon dioxide, NAO is the North Atlantic oscillation, and IPO is Interdecadal Pacific oscillation. f) Lead-lag relations LL(IPO, SAOD) and LL(NAO, SAOD). The range for the LL- values are -1, +1, but for the LL(IPO,SAOD) we have multiplied the range with 0.5 to allow comparison of the two LL series.

**4.2.2 Cycle periods**

The cycle periods for the five time series are calculated with the cumulative angle method. In Fig. 4a we apply and compare the two methods for calculating cycle periods for $CO_2$. The calculations show that the cumulative method and the trace method give roughly similar developments of the cycle periods. Droplines show that the periods (the series amplitude) are relatively shorter, $\approx 188 \pm 68$ years, around volcanic eruptions than between eruptions. The thin dark yellow line around the 1253 eruption shows the high- resolution cycle periods. Calculation and results for the other alternative variables are shown in Fig. 4b.

A PCA plot for the cycle periods of the five time series: $CO_2$, NAO, IPO, SAOD and NHST are shown in Fig. 4c. (The TSI series is too short to be included). Some relevant OLR's are:

$$CPIPO = - 0.262\ CPSAOD + 75.7, R^2 = 0.277, p < 0.001, n = 1994 \qquad (5)$$
$$CPNAO = -1.031\ CPIPO + 135.9, R^2 = 0.11, p < 0.001, n = 1994 \qquad (6)$$
$$CPCO_2 = - 0.234\ CP(NHST) - 38.1, R^2 = 0.07, p < 0.001, n = 1850 \qquad (7)$$

The cycle periods for IPO are leading SAOD most of the time after the year 500 (black bars in Fig. 4d), whereas NAO is lagging SAOD most of the time (red bars). Thus, most of the time $\approx 500$ to 1850 we find that cycle periods show LL relations as IPO $\rightarrow$ SAOD $\rightarrow$ NAO. Exceptions are the periods following the 536 eruption and the 1230 eruption that affect LL relations for the ocean variable relative to SAOD. The range for the bars is -1 to +1, but to compare the two LL relations we let the black series range from -0.5 to +0.5.

We now have twenty time series available for assessing the possible (partial) causal role of volcanism on the two ice ages. i) the five time series, NHST, SAOD, TSI, NAO, IPO and $CO_2$. ii) five time series that show cycle periods for the series in (i). Ten time series that show the LL relations between the series in (i). To these series we examine relations between them by applying the six tools developed in the method section: OLR, PCA, LL relations, cooling characteristics, events that are fixed in time, and physical explanations. The first four tools allow calculation of confidence intervals; the last two tools are qualitative.





440 Table 2 Candidate variables for extending the ice ages. Cold and decreasing periods are First millennium: 1-41, 245-626, 685-

884, 980-1000; Second millennium: 1039-1330,1480-1824.

| Variable | Acronym | Series period | Cooling & Leading | Cycle period, Cum. Method, years | Time series source |
|---|---|---|---|---|---|
| Northern Hemisphere summer temperatures | NHST | 1-2017 | n.a. | Cum:79.1±21.1 N=19 Trace: 73.9±39.2 [1] Mode:63-≈200 | Buntgen et al. (2020) |
| Volcanic eruptions | SAOD | 1-2015 | 654, 51% | Cum: 97.5±53.9 N =15, Trace: 68.8±42.5 [2] Mode: 62 | Sigl et al. (2015) |
| Total solar insolation | TSI-17 [3] TSI-5 TSI L(0.2) | 1700-2022 | 23, 1.7% | Cum:12.2 ±10.3 Cum:21.1±19.8 Mode:7 Cum 74.5±24.4[4] | Clette et al. (2015) |
| Carbon dioxide | $CO_2$ | 1-1850 | 207, 16% | Cum 203.7±98.3, N=9 Trace: 222.1±164.3 Mode: 112-257 | Neukom et al. (2019) |
| North Atlantic oscillation | NAO | 1-2010- | 139, 11 % | Cum:89.4±30.3 N=22 Trace73±31.3 Mode 62 | Hernández et al. (2020) |
| Interdecadal Pacific oscillation | IPO | 1-2022 | 356, 28 % | Cum:70.1±23.5 N=28 Trace:60.9±24.2 Mode: 52 | Vance et al. (2022) |

(1) NHST. A very long cycle period from year 1 to 590 is removed. The distribution is bimodal with peaks at ≈ 63 and 200 years.

(2) SAOD. The very long cycle period at year 604 is removed. The distribution has a peak at 61 years.

445 (3) The time series for TSI were shifted 17- and 5-time steps back respectively.

(4) TSI 11 years average LOESS(0.2) smoothed





**Figure 4** Cycle periods and LL relations. a) Two calculations of running cycle periods for CO2. Red curve shows LOESS(0.3) smoothed running estimate calculated by the cumulative angle method. Blue line shows the traced closed curve in phase space method. Thin, dark yellow line show a high-resolution version of the smoothed cycle periods. Droplines show large volcanic emissions (<- 10Wm⁻²). b) Running cycle periods for the five time series, CO2, NAO, IPO, SAOD and CO2. Traced closed curves method. Droplines show the two large volcanic eruption in 536 and 1230. c) PCA loading plot for the cycle periods for the five timeseries. d) LL relations between NAO and SAOD (red bars) and between IPO and SAOD (black bars). Droplines show the two large volcanic eruptions in 536 and 1230



**5.  DISCUSSION**

Our aim is to determine if the causal relations between volcanism, alternative cooling mechanisms, and ice ages can be strengthened. In the discussion, we show results that both strengthen and weaken causal relations between variables.

**5.1 Volcanic eruptions and NHST**

We apply the LL algorithm to the SAOD and NHST variable pair and to the CP(SAOD) and CP(NHST) pair. When the
SAOD time series leads the NHST, volcanic eruptions have the potential to affect NHST. Below, we identify such periods as SAOD effect periods.

The formal definition of the LIA is from 1250, but there was a large volcanic eruption in 1230 (global forcing -15.9-watt $m^{-2}$). From 1184 to 1417, and including two major eruptions, SAOD is leading NHST. There is a second long period from 1619 to 1758 where SAOD leads NHST and during which there were several significant eruptions.

*Support for hypothesis, H1*. Our results partly support our first hypothesis, **H1**, that effects of major volcanic eruptions expressed by the SAOD could lead NHST and decrease temperatures during the two ice ages. The negative correlation coefficient, R = - 0.56, that we calculated for SAOD = f(NHST) (the LOESS(0.2) smoothed version) corresponds well with the significant negative correlation of R =- 0.56 reported by Buntgen et al. (2020) between NHST and SAOD (Büntgen use the same time series, but not LOESS (0.2) smoothed). Both SAOD and NHST showed cycle periods around 60 to 70 years
and their cycle period distribution showed a peak in the CP histogram at 62- 63 years, but also peaks at longer periods (Fig. 2f). Since it is the SAOD that can impose cycle periods on the NHST, the results suggest that SAOD contributed a partial causal effect to the NHST.

*Issues that should be resolved*. However, a long run decline in temperature occurred from year 245 and before the formal definition of the LALIA and the twin large volcanic eruptions, Fig. 2c. A short SAOD effect that lasted 53 years from the year
327 to 379 and decreased the NHST. The next SAOD effect lasted from 438 to 529, and thus ceased 7 years before the first of the twin large volcano eruptions in 536 and 540. In addition, there were periods before the LALIA that did not have SAOD as a leading variable. Last, and in contrast to our hypothesis, the series for cycle periods did not show that cycles in the SAOD series were persistently leading cycles in the NHST series.

**5.2 Alternative cooling variables**

We discuss four alternative candidate variables that may continue decreasing or maintaining the cold temperature. The signs added to the variable acronyms indicate if low or high values of the variable contribute to cold NHST, and the percentage shows the contribution of the variables to cooling during the ice ages.

*Total solar irradiance*, TSI (-, 2%). The results for TSI showed that there is a close correspondence between TSI and NHST from 1700 to present, Fig. 3a. The correspondence seems to be valid both for short (≈ 30 - 50 years) and for longer centennial



cycles. Variation in solar intensity is of the order of 1.5 Wm$^{-2}$ (Vieira et al., 2011, p.17) and thus too small to account for the variation in temperature on its own. Other variables may covary with TSI and enlarge the effects of TSI (Stoffel et al., 2022, p. 1084). The TSI shows two cycle periods, 11 and 22 years for the 11 years averaged series that is established for TSI, (Steinhilber et al., 2009, p. L19704), but 74.5± 24.4 years cycle periods for the LOESS(0.2) smoothed series. Thus, TSI could function as control knobs for climate variables. Hernández et al. (2020, Fig2) show for example, that small numbers of

sunspots correlate with low NAO values (≈ years 600- 2000).

*Ocean variables NAO (-, 11%), IPO (+, 28%).* During the Common Era, NAO and IPO changed, but also overlapped, in being a leading variable to SAOD. For the series that characterize the North Atlantic waters in addition to NAO, (the Atlantic meridional overturning circulation (AMOC) and the Atlantic multi-decadal oscillation (AMO) their LL relations are NAO → AMO → AMOC for the period 1880 to 2020 (Seip & Wang, 2022). All the North Atlantic variables could cause cooling of

the NHST, but probably through different mechanisms (Only AMO is measured in temperature units.). van Dijk et al. (2022, pp. 1601, 1607) suggest, with reservation, that an increase in AMOC that peaked 10 years after the LALIA (536 /540) contributed to preserving the surface cooling. The NAO and IPO both show cycle periods ≈ 70- 90 years and both have a standard deviation, SD, of about 30%. During the LIA, and in contrast to the situation for the LALIA, NAO was leading SAOD and IPO was lagging SAOD from about 1400 to 1500, but the roles switched from 1500 to about 1850.

The periods may be generated internally, Arzel et al. (2018); by stochasticity in the interaction between ocean basins, Seip and Grøn (2018); determined by the physical interaction between the Atlantic and the Pacific (Meehl et al., 2021), by shifts between cold and warm eras (Alonso-Garcia et al., 2017; Stevens et al., 2016), by sea-ice /ocean feedbacks ≈1200 to ≈ 1900 (Miller et al., 2012 p. L02708); by volcanism or by external forcings.

*Carbon dioxide, CO$_2$.* (-, 16%) Volcanic eruptions instigated directly or indirectly increases in atmospheric CO$_2$

concentrations for 136 ±42 years, Fig. 3b. This increase could counteract the increase in NHST caused by aerosols. The sources for the increase in atmospheric CO$_2$ emissions from subaerial volcanic regions, Werner et al. (2019, Fig 8.3) and Buono et al. (2023), and emissions from oceans in the Southern Hemisphere that get warmer. Figure 3b shows that atmospheric CO$_2$ increases after volcanic eruptions and Fig. 4a (dark yellow line segment from 1230 to 1319) suggests that volcanic eruptions lead to a short cycle period in CO$_2$ .

**5.2.1 The strength of alternative variables**

We have shown that the alternative variables lead NHST, and their mode is in a state that would cool the Northern Hemisphere, but they should also have the strength to give a significant impact on the temperature. Wu et al. (2019) suggest that AMO and PDO variabilities contribute 30% to the variability of global temperature anomalies. The IPO cycle time was similar to the cycle times found by Vance et al. (2022) of 61± 56 years and by Yang et al. (2020 p.1195) for global cycles

(30°S, 60°N) in sea surface temperature, SST, on multidecadal timescales. Decreasing temperatures caused by ocean variabilities or other climate variables could compensate for the large temperature increase caused by postindustrial emissions





of $CO_2$ until 2008 -2010, Seip and Wang (2023, Fig 4e) and Buntgen et al. (2020, p.5). Dong and McPhaden (2017) suggest that both IPO and volcanic eruptions caused the hiatus in global warming during the 1946 -1976 hiatus period and similar co-occurring events may have acted during the ice-age periods. Volcanic eruptions and subaerial volcanic regions are estimated

to emit 540 Tg $CO_2$ yr$^{-1}$, (Werner et al., 2019, p. 194). This is about 1% of the current annual increase in $CO_2$, (37.10$^6$ Tg $CO_2$ yr$^{-1}$, 2021), but a much higher proportion of the maximum $CO_2$ load during the preindustrial time (Seip & Wang, 2023). Thus, both the timing and the strength of ocean variabilities, in particular the IPO that counts for almost 50 % of the "void" periods, would be sufficient to maintain and cool the NHST.

Among the auxiliary variables, IPO was the mechanism that contributed most to NHST cooling during periods when
volcanism did not cool, Fig. 3e. After the twin eruptions in 532/540, CPNAO and CPIPO were leading CPSAOD. This would contrast with SAOD periods as instigating long cycle periods in NAO or IPO. However, after the volcanic eruptions in 1230 and 1258, CPSAOD was leading CPIPO, Fig. 4d.

*Support for hypothesis H2.* We found that the ocean variable IPO (+, 28%), NAO (-,11%) and atmospheric $CO_2$ (-, 16%) could replace volcanism as cooling mechanisms during periods when effects of volcanism ceased. Our results showed that IPO
was the variable that filled most "voids" as cooling variable, Figure 3e, and when SAOD showed short cycles corresponding to the short duration of volcanic eruptions, IPO showed long cycles, Fig 4c.

*Issues that should be resolved.* Since the ice ages last for 124 and 600 years respectively, and oceans show cycle periods of 40 to 80 years, Figure 4b, the oceans change in being in cooling and warming modes during the ice ages. Since there is an overall cooling during the ice ages, some mechanism must secure that the cooling persists.

The rapid increases in $CO_2$ instigated by volcanic eruptions, Fig. 3b and Fig. 4a, suggest that volcanic eruptions also could increase global temperature. However, on the balance, the cooling effects from oceans seem to outweigh the warming effect from atmospheric $CO_2$.

**5.3 Does volcanism instigate secondary cooling mechanisms, or do they occur by chance?**

Several studies show that ocean variabilities are globally connected across ocean basins (Seip et al., 2023; Yang et al.,
2020), but the mechanisms that govern those associations are not well known. The LL relation between variabilities changes direction over decadal time scales, but volcanism does not seem to impose such changes, Fig 3f. On the other hand, volcanism may instigate a change in cycle periods in ocean variability series by cooling ocean waters, Fig. 4b. We do not know if such cooling events will change an underlying variability, or if it just adds an extra cycle period that is superimposed on the underlying variability.

Volcanism may change sea surface temperatures and thereby change sea level pressure (SLP) and the North Atlantic oscillation mode (NAO), van Dijk et al. (2022, p. 1610), and then continue the cooling effect. A second effect of cooling of the oceans is that it would also alter the exchange of $CO_2$ between the oceans and the atmosphere, DeVries (2022) and Gruber



et al. (2019 b). A third effect of the cooling instigated by volcanism could be a terrestrial response as a decrease in $CO_2$ caused by an increase in net ecosystem productivity, (Jones & Cox, 2001, modeling study).

*Support for hypothesis H3.* Our third hypothesis, **H3**, that volcanic eruptions would instigate alternative cooling variables could be partially supported. IPO are closely associated with the "void" time periods and when the cycle periods for SAOD are short, the cycle periods for IPO are long, Fig. 4c. However, to strengthen the association, there are some additional criteria that should be met.

     *Issues that should be resolved.* We must show that SAOD is leading IPO during the transition from cooling caused by 580 volcanism to cooling caused by IPO. There are weakly indications that this is the case (Fig. 3f and Fig. 4d). The contrasting hypothesis, that the alternative mechanisms are in cooling modes by chance, also has some support. *First*, there is a relatively small standard deviation of 33-34% for the cycle periods NAO and IPO suggesting that there is more regularity in their positive and negative modes than volcanic eruptions would indicate. *Second*, there are several alternative explanations for a high degree of stability in ocean variabilities, e.g., Di Lorenzo et al. (2023). *Third*, if the ocean variabilities change in cooling the NHST, 585 it follows from the relatively regular cycle periods, that they also change in warming the NHST. We do not know if cooling or warming the surface temperature by the oceans have different strengths, but we did our calculations by adopting the usual procedure of separating the variabilities in two halves (+ warm and – cold) based on a linear regression for the data. Miller et al. (2012) and Alonso-Garcia et al. (2017), although with a little different angle, suggest that there is a dynamic interaction between warm and cold periods mediated by melting and rebuilding of glaciers and sea ice in the Arctic on a centennial scale. 590 However, on the preponderance, we believe that alternative cooling variables were instigated by volcanic eruptions.

**5.5 Robustness**

     There are three major caveats related to our results. *First*, the time series we use are based on proxy records for SAOD, NHST, NAO and $CO_2$. The series may be biased both in amplitude and timing, *Second*, disentangling series has no canonical solution, because patterns in the observed series may be generated by the interaction between series that represent different 595 mechanisms. *Third*, volcanism has several attributes that could affect its effectiveness as a "cooling machine." For example, Jenkins et al. (2023), discussing the Tonga eruption in January 2022, suggest that because it injected a much larger amount of water vapor ($H_2O$) than sulfur ($SO_2$) into the stratosphere, the net effect could be that the global surface temperature would increase.

     The robustness of the method can be evaluated by applying it to sine series that are shifted in time. In Fig. 2d we show how 600 the CP is identified in a sine function with variable cycle periods. However, an application to observed series may be a more true-to-life test. Seip et al. (2019) applied it to an evaluation of leading indexes used for forecasting Industrial production (IP) in German economics. The result showed that the forecasting series were correctly leading IP – and therefore successful - 80% of the time.




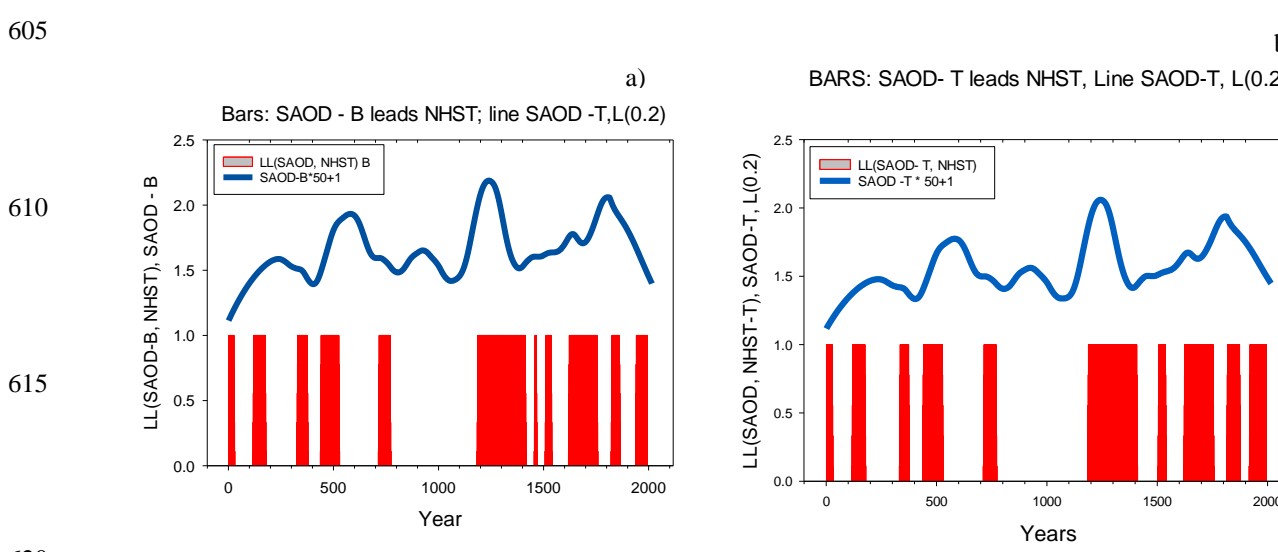


**Figure 5**. Comparing LL relations for LL(SAOD-B, NHST) and SAOD -T where SAOD – B is the SAOD series used by Buntgen et al. (2020) and SAOD – T is the revised series provided by Toohey, (Aubry et al., 2020). The blue line is the LOESS(0.2) smoothed NHST series


When it was not successful, the economy was anomalous. The running cycle periods algorithm can be evaluated by comparing Fig. 3b, the $CO_2$ time series and Fig. 4a, which shows that the short cycle periods prior to large eruptions are reproduced well. Alternative LL methods exist, like cross correlation (CC) techniques to determine if one cyclic series is a candidate cause for a second series, but with a time lag (Kestin et al., 1998). The CC method presupposes that the two cyclic series are (semi-) stationary and with several, $\approx 7$, cycle periods.

## 6. CONCLUSION

We identified the periods during the Common Era with decreasing or cold temperatures in the Northern Hemisphere to last for 1282 years. We found the cold periods caused by volcanic eruptions (as SAOD) to last for 659 years, 51 % of the cold years based on a leading role for SAOD to NHST.

The duration of cooling by TSI was 2 %, NAO, 11%, IPO, 28% and $CO_2$, 16 % of the time. However, the cooling periods for the last four climate variable overlap, so the net percentage of time the five climate variables cool the Northern Hemisphere is 89 % of the time. Ocean cooling (NAO, IPO) and $CO_2$ coincide in cooling time for 319 of the 495 years when NAO and IPO are in cooling modes.

We found that the running cycle periods for NAO and IPO showed a standard deviation of 33-34 % around their respective means of 89 and 70 years and were of the same order as the cycle periods for NHST (74 to 79 years) and volcanic eruptions



(69 to 98 years). However, we identify IPO only cursory as a lagging variable to SAOD. Still, we conclude that volcanism can be, on the preponderance, an instigating force for variables that maintain or contribute to persistent cooling. However, we also found an opposing effect, volcanic eruptions were associated with short cycle periods of increasing atmospheric $CO_2$ concentrations.

To explore relations between volcanism and Ice-ages further, the dynamics that create similar cycle periods in ocean variability series and volcanism should be studied further. This could be done with simulation models where simulated time series are compared to lead-lag series and cycle period series in the present study.





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





## Code availability

All calculations are available from the first author. The essential calculations are made in Excel, on one Excel sheet

## Data availability

All data are available from the first author

## Interactive Computer environment

An Excel Book is added as a Supplement.

## Author Contributions

KLS and ØG  designet the study and KLS wrote the first draft. KLS and ØG developed the calculation method and KLS did the calculations. ØG and HW controlled the scientific  information. KLS wrote the final manuscript with contributions from both co-authors.

## Competing interest

The authors declare that they have no conflict of interest.

## Color shemes

The first author ar partially colour blind, but distinguish the colours in the Figures