# Peer review of "Identifying climate variables that interchange with volcanic eruptions as cooling forces during the Common Era's ice ages."

_EGUsphere, 2024_

## Author Comment (AC2)

**Identifying climate variables that interchange with volcanic eruptions as cooling forces during the Common Era's ice ages**

1. From referee #1. The claim on lines 529-530 that "Volcanic eruptions instigated directly or indirectly increases in atmospheric $CO_2$ concentrations for 136 ±42 years" is completely wrong and based on statistical artifacts of the analysis that have no basis in physics.

The text that the referee refers to is on lines 526 -531:

*Carbon dioxide, $CO_2$*. (-, 16%) Volcanic eruptions instigated directly or indirectly increases in atmospheric $CO_2$ concentrations for 136 ±42 years, Fig. 3b. This increase could counteract the increase in NHST caused by aerosols. The sources for the increase in atmospheric $CO_2$ emissions from subaerial volcanic regions, Werner et al. (2019, Fig 8.3) and Buono et al. (2023), and emissions from oceans in the Southern Hemisphere that get warmer. Figure 3b shows that atmospheric $CO_2$ increases after volcanic eruptions and Fig. 4a (dark yellow line segment from 1230 to 1319) suggests that volcanic eruptions lead to a short cycle period in $CO_2$. (Figure 4 in main text)

We could have been more careful, for example written: "Volcanic eruptions appear to instigate directly or indirectly increases in atmospheric $CO_2$ concentrations with time lags from zero to 40 years. We have not identified conclusively the sources for the increases in $CO_2$"

[Figure]

Figure 3 b in the manuscript is shown to the left. The right figure shows $CO_2$ and large volcanic eruptions. The eruptions tend to be followed by increases in $CO_2$, but some with lags 10-40 years.

2. From referee #1: The conclusion that $CO_2$ changes are as important as volcanic eruptions is ridiculous, if you account for the magnitude of their forcings, except for the period since 1850 when $CO_2$ and other greenhouse gases dominate.

Our text starting on line 268: The lower blue horizontal lines show periods with low temperature and the horizontal upper green lines show periods with decreasing temperature. The red bars show when SAOD is leading NHST. Altogether, there are seven periods lasting on average 75 ± 32 years (range 10 to 96 years) before and during the LALIA where other variables decrease or maintain low temperatures.

We could have been more careful and written: "Over short periods, ≈ 2 years, large volcanic eruptions, e.g., <-10Wm$^{-2}$, may cause dramatic cooling. However, over longer decennial time spans decreases in $CO_2$ may cause volatile, but persistent decreases in NHST during the Common era, but before the start of the industrial revolution.

[Figure]

Figure 2 c in the manuscript is shown to the left. The upper left green line shows a period with decreasing temperature. The right figure shows the downward sloping trend corresponding to the horizontal green line.

3. From referee# 1: "By treating SAOD, TSI, NAO, IPO, and $CO_2$ as normalized time series, and not with units and actual radiative forcing, the results are meaningless, particularly as NAO and IPO are internal variability and not forcing"

The NAO is measured as air pressure. It is normal to refer to the NAO as being in positive or negative phase. See figure from NOAA below. The target variable for the study is the Northen hemisphere summer temperature and the conversion from NAO in air pressure units to temperature in the Northern hemisphere is complex.

We think statistical-and difference modeling complement each other. The first method require careful evaluation of possible cause - effect results (cross validation would secure that the cause comes before the effect), and centered and normalized data are frequently used in that context. Difference modeling require much more data that often will consist of non-linear relations, e.g., Seip (1991).

NAO time serieshttps://psl.noaa.gov/data/timeseries/daily/NAO/

**North Atlantic Oscillation (NAO)**

**Description:**
Pattern of variability in the North Atlantic.
**Image:**

[Figure]

Postive Phase:

**Calculation Method:**
The indices are based on centers-of-action of 500mb height patterns. These time series utilize the NCEP/NCAR R1 dataset. The area averaged region 55-70N;70W-10W is subtracted from 35-45N; 70W-10W. The 1981-2010 period was used as climatology. Before computing these indices the 500mb height fields are spectrally truncated to total wavenumber 10 in order to emphasize large-scale aspects of the teleconnections.
**Time Interval:** Daily
**Time Coverage:** 1948-near present

References

Buono, G., Caliro, S., Paonita, A., Pappalardo, L., & Chiodini, G. (2023). Discriminating carbon dioxide sources during volcanic unrest: The case of Campi Flegrei caldera (Italy). *Geology*, *51*(4), 397-401. https://doi.org/10.1130/G50624.1

Seip, K. L. (1991). The ecosystem of a mesotrophic lake- I. Simulating plankton biomass and the timing of phytoplankton blooms. *Aquatic Science*, *53*(2/3), 239-262. https://doi.org/1015-162/191/030239-24

Werner, C., Fischer, T., Aiuppa, A., Edmonds, M., Cardellini, C., Carn, S., & Allard, P. (2019). Carbon Dioxide Emissions from Subaerial Volcanic Regions: Two Decades in Review. In B. Orcutt, I. Daniel, & R. Dasgupta (Eds.), *Deep Carbon: Past to Present* (pp. 188-236). Cambridge University Press.

---

## Author Comment (AC3)

**Response to referee# 2 **Identifying climate variables that interchange with volcanic eruptions as cooling forces during the Common Era's ice ages.**

**Referee # 2**

The authors have made a commendable effort in this manuscript....

Response. Thank you for the kind assessment.

**Referee #2**

However, the clarity of the manuscript should be improved.

Response: We agree. We will make efforts in making the text easier to read.

**Referee #2**

The first line of the discussion is actually helpful but should be moved up to the introduction.

Response:

Agree, the sentence is moved up to the introduction.

**Referee #2**

"… my primary concern lies with the tree-ring width (TRW) reconstruction, which forms the foundation of the study."

." … but they fail to mention that the reconstruction is based on tree-ring widths, which are known to exhibit biological memory effects (see Esper et al., 2015, among many other references). This omission is critical, as memory effects in TRW can extend the cooling signal by up to 10 years, whereas the actual volcanic forcing and cooling feedback may be much shorter in duration."

Response:

Thank you. The first part of the sentence in the abstract "… little is known about when the effects of volcanism ends, and which other mechanisms prolong… … "(line 8) and a similar sentence in the introduction (line 25), underestimate the current knowledge of the cessation of cooling from volcanic eruptions, e.g., Esper 2015 paper, page 66. The sentences are reformulated to acknowledge the present knowledge of volcanic cooling durations.

**Referee #2**

Therefore, the paper could benefit from using Northern Hemisphere reconstructions based on maximum latewood density (MXD), such as those by Schneider et al. (2017) or

Büntgen et al. (2024), as MXD reconstructions are known to mitigate the memory effects associated with TRW.

Response:

We use two attributes of the data, i) their positive (+) and negative (-) values defines as the values (+) above the average and the values below (-) the average and ii) their LL relations to the SAOD time series.

As shown by the referee, the NHST based on tree ring data overestimate the duration of the cooling from volcanic eruptions (Schneider et al., 2017). Unfortunately, the data are only available for the period 1001 to 2023. (There are series that are based on both MXD data and tree-ring data, Schneider et al. (2017, page 3), but here we only examine the MXD data series.) Figure 1a compares NHST time series based on the tree-ring data to those based on the MXD data.

Figure 1b shows when SAOD is leading NHST(MXD). For much of the millennium, the leading role of SAOD to NHST(MXD) is quite good. The volcanic eruptions and the decreases in temperature following the eruptions are from Tejedor et al. (2021, p. 7), but the study list only eruptions before 1835. Our LL results suggest that there should be an eruption around 1930, and it could be the Stromboli 1930 eruption, but although the 1930 eruption was strong, it show persistent eruptions over long time spans.

**Referee #2**

The authors apply a loess filter to the data (what happens if you use the raw data?),

[Figure]

Figure 1. NHST based on tree-ring data and on MXD data. a) Comparing NHST data based on tree-ring data and on MXD data for the period 1001 to 2022. b) LL relations between SAOD and NHST (MXD). The droplines show volcanic eruptions during the last millennium up to 1835. The blue dots show the temperatures just after the eruptions or

one year later. The red columns show the periods where the effects of the eruptions as SAOD leads NHST(MXD). Data are from Tejedor et al. (2021).

Unfortunately, we do not have NHST(MXD) data for the first millennium of the Common Era. However, those data would have more uncertainty added due to decreased proxy availability, Tejedor et al. (2021, p. 11).

**Referee #2**

The authors apply a loess filter to the data (what happens if you use the raw data?)

**Response**

Since we study long term temperature variations, we use the LOESS (0,2) smoothed data. Using raw data, the picture would be as in Figure 2.

[Figure]

Figure 2. Same as in Figure 1, but with the raw NHST(MXD) data.

Most of the data are contaminated with noise and only a few are significant with LL> 0.32.

**Referee #2**

In Figure 3, if you include the titles (which i dont know if its allowed), you should also point out which are your NHST raw temperature and which are the PCAs.

**Response**

Thank you. We forgot to give a text for panel e) The panel shows the PCA is applied to the LOESS(0.05) smoothed values.

References

Schneider, L., Smerdon, J. E., Pretis, F., Hartl-Meier, C., & Esper, J. (2017). A new archive of large volcanic events over the past millennium derived from reconstructed summer temperatures (vol 12, 094005, 2017). *Environmental Research Letters*, *12*(11). https://doi.org/ARTN 119501

10.1088/1748-9326/aa9426

Tejedor, E., Steiger, N., Smerdon, J. E., Serrano-Notivoli, R., & Vuille, M. (2021). Global Temperature Responses to Large Tropical Volcanic Eruptions in Paleo Data Assimilation Products and Climate Model Simulations Over the Last Millennium. *Paleoceanography and Paleoclimatology*, *36*(4). https://doi.org/ARTN e2020PA004128

10.1029/2020PA004128